# Usefulness of Staging Chest CT in Breast Cancer: Evaluating Diagnostic Yield of Chest CT According to the Molecular Subtype and Clinical Stage

**DOI:** 10.3390/jcm10050906

**Published:** 2021-02-25

**Authors:** Seulgi You, Tae Hee Kim, Doo Kyoung Kang, Kyung Joo Park, Young-Sil An, Joo Sung Sun

**Affiliations:** 1Department of Radiology, Ajou University School of Medicine, Suwon 16499, Korea; seulgi88322@gmail.com (S.Y.); taehee78@hanmail.net (T.H.K.); kdklsm@ajou.ac.kr (D.K.K.); kjpark@ajou.ac.kr (K.J.P.); 2Department of Nuclear Medicine and Molecular Imaging, Ajou University School of Medicine, Suwon 16499, Korea; aysays77@naver.com

**Keywords:** staging chest CT, breast cancer, diagnostic yield, molecular subtype, clinical stage

## Abstract

The aim of this study is to investigate the clinical utility of staging chest CT in breast cancer by evaluating diagnostic yield (DY) of chest CT in detection of metastasis, according to the molecular subtype and clinical stage. This retrospective study included 840 patients with 855 breast cancers from January 2017 to December 2018. The number of patients in clinical stage 0/I, II, III and IV were 457 (53.5%), 298 (34.9%), 92 (10.8%) and 8 (0.9%), respectively. Molecular subtype was identified in 841 cancers and there were 709 (84.3%) luminal type, 55 (6.5%) human epidermal growth factor receptor 2 (HER2)-enriched type and 77 (9.2%) triple-negative (TN) type. The DYs in clinical stage 0/I, cII, cIII and cIV were 0.2% (1/457), 1.7% (5/298), 4.3% (4/92) and 100.0% (8/8), respectively. The DYs in luminal type, HER2-enriched type and TN type were 1.7% (12/709), 3.6% (2/55) and 2.6% (2/77), respectively. Clinical stage was associated with the DY (*p* = 0.000). However, molecular subtype was not related to the DY (*p* = 0.343). Molecular subtype could not provide useful information to determine whether staging chest CT should be performed in early-stage breast cancer. However, chest CT should be considered in advanced breast cancer.

## 1. Introduction

Breast cancer is the most common cancer among female patients, accounting for 30% of all cancers and 14% of all cancer-related deaths in female patients [1]. Initial distant metastasis in newly diagnosed breast cancer can adversely affect patient prognosis. The five-year survival rate of breast cancer patients with initial metastasis is about 25%, which is much lower than that of patients without initial metastasis (up to 80%) [2,3]. Therefore, precise preoperative staging is very crucial to determine a proper therapeutic plan.

The prevalence of distant metastasis in early-stage breast cancer was as low as 0.2% for stage I and as 1.2% for stage II cancers [4,5]. Most of the existing guidelines, including those proposed by the National Comprehensive Cancer Network (NCCN), European Society for Medical Oncology (ESMO), and American Society of Clinical Oncology (ASCO), do not recommend routine use of preoperative chest CT in early breast cancer staging for detection of distant metastasis [6,7,8]. Nevertheless, many asymptomatic patients with breast cancer undergo chest CT for evaluation of distant metastasis [9]. The guidelines recommend chest CT in the initial staging workup for cases showing clinical signs, symptoms, or laboratory values suggesting the presence of metastases, clinically positive axillary nodes, large tumors, or aggressive biology; however, the relevant criteria are somewhat ambiguous [6,7,8,10]. Simos et al. reported that one of the potential factors that may influence the physician’s decision to perform preoperative chest CT is an aggressive molecular subtype such as human epidermal growth factor receptor 2 (HER2)-enriched type or triple-negative (TN) type [11]. Molecular subtype of breast cancer could be an important factor in deciding whether to perform initial staging chest CT.

Invasive breast cancer is classified into four subtypes on the basis of its molecular characteristics: luminal A tumors, luminal B tumors, human epidermal growth factor receptor 2 (HER2)-enriched tumors, and triple-negative (TN) tumors. Luminal-type breast cancers are characterized by the expression of both estrogen receptor (ER) and progesterone receptor (PgR) [12]. Luminal breast cancers show better prognosis and better response to hormone receptor-targeted therapies [13]. HER2-enriched cancers overexpress the HER2/new gene, while TN breast cancers do not show hormone receptors or HER2 overexpression [12]. HER2-enriched and TN cancers are more aggressive and have poorer outcomes than luminal-type cancers [14]. These tumors show a higher rate of locoregional recurrence and lower survival rate after distant metastasis [15,16,17,18]. Since the prognosis of breast cancer differs according to the molecular subtype, guidelines recommend core biopsy for immunohistochemical assessment of ER, PgR, and HER2 status to classify the molecular subtype of breast cancer during initial diagnosis [6,7,8]. We could infer that the molecular subtype of breast cancer might affect the initial metastasis of breast cancer. However, there is no information regarding the prevalence of initial distant metastasis and the benefits of initial chest CT in detecting distant metastasis based on the molecular subtype of breast cancer.

Therefore, in this study, we aimed to investigate the benefit of chest CT in initial staging by evaluating the diagnostic yield (DY) on the basis of the molecular subtype and clinical stage of breast cancer. The usefulness of chest CT can be evaluated in terms of diagnostic yield (DY), which indicates the proportions of patients with true-positive metastases and false-positive metastases.

## 2. Materials and Methods

Our institutional review board approved this retrospective observational study and waived the requirement for informed patient consent.

### 2.1. Study Design and Patients

The current observational study was undertaken retrospectively using a single-center cohort. We reviewed consecutive data of 1215 patients initially diagnosed as having breast cancer (18 patients with bilateral breast cancer) from January 2017 to December 2018 at *** University Hospital, a tertiary referral center. The exclusion criteria were as follows: (1) no initial staging chest CT examination (*n* = 347) and (2) a history of malignancy other than breast cancer with the potential to metastasize (*n* = 13). Finally, we included 840 patients with 855 breast cancers (15 cases of bilateral breast cancer) for analysis.

### 2.2. Image Acquisition

All chest CT examinations were performed using a 64- or 32-channel scanner (Brilliance 64; Philips Healthcare, Cleveland, Ohio, or Aquilion ONE, Canon medical systems, Japan). The CT scans were obtained in the end-inspiration state, as far as possible, in the supine position. Chest CT scans were performed from the thoracic inlet to the upper abdomen covering the entire liver. Post-contrast chest CT images were obtained using nonionic contrast media (2.5 mL/s, Iomeprol—Iomeron 300; Bracco, Milan, Italy) with a fixed delay of 45 s. Scanning parameters were as follows: tube voltage, 120 kVp; tube current, 200 mAs with automatic tube current modulation; tube rotation time, 0.5 s; pitch, 0.938; collimation, 16 × 0.75 mm; slice thickness, 3 mm.

### 2.3. Clinical and Image Analysis

#### 2.3.1. Analysis of Breast Imaging

We reviewed all patients’ medical records for clinical and image analysis. Patients’ clinical stage was determined according to the eighth edition of the Tumor-Node-Metastasis (TNM) based staging system of the American Joint Committee on Cancer (AJCC) [19]. The clinical stage of breast cancer was determined based on clinical information and the results of breast ultrasound (US) and breast magnetic resonance imaging (MRI) by any of three board-certified breast radiologists (with 20, 12, and 2 years of experience) as a part of daily practice.

#### 2.3.2. Analysis of Chest Imaging

Two chest radiologists (19 and 3 years of experience), one of whom did not participate in the evaluation of staging chest CT scans, reviewed reports of staging chest CT based on the routine daily practice of board-certified chest radiologists (30, 19, and 3 years of experience) to identify the presence of distant metastasis in lung, liver, bone, and lymph node (LN). They classified the results using the following four-point scale for the presence of metastasis: 1, no metastasis; 2, probably no metastasis; 3, probable metastasis; 4, definite metastasis. For pulmonary metastasis, only solid nodules were regarded as metastasis because sub-solid nodules rarely indicate pulmonary metastasis in breast cancer [20,21]. For lymph node metastasis, supraclavicular lymph nodes with a short-axis diameter of ≥5 mm on staging chest CT were interpreted as positive for metastasis [22,23]. For the mediastinal and hilar regions, metastatic lymph nodes were defined as having a short-axis diameter ≥10 mm on staging chest CT [24]. For hepatic metastasis, lesions showing hypoattenuation on unenhanced CT and enhancement less than the surrounding liver on contrast-enhanced CT were regarded as metastasis [25]. Skeletal metastasis was indicated by osteolytic, osteoblastic, or mixed lesions of the thoracic skeleton in the bony thoracic cage, including the vertebral bodies, ribs, scapulae, and clavicles [26,27]. In addition, important findings other than breast cancer metastasis on staging chest CT scans were marked for follow-up.

### 2.4. Reference Standard

True metastasis was established when the findings met one of the following criteria; (1) the lesion was pathologically confirmed as metastasis, (2) the lesion progressed at subsequent follow-up CT examinations, (3) the lesion decreased in size after chemotherapy, consistent with the response of other primary or metastatic lesions, (4) the lesion showed hypermetabolism in fluorine-18-fluorodeoxyglucose positron emission tomography (FDG-PET) CT, or (5) the lesion showed consistent findings in subsequent imaging studies, including MR or bone scans. Benign lesions were confirmed when the findings met one of the following criteria: (1) the lesion was pathologically confirmed as benign, (2) the lesion did not change in size for at least one year, (3) the lesion decreased in size or disappeared without any treatment on follow-up CT, (4) additional FDG-PET scan showed findings consistent with benign lesions, or (5) the lesion showed consistent findings in subsequent imaging studies, including MR or bone scans.

### 2.5. Statistical Analysis

On the basis of these findings, to assess the usefulness of chest CT-based staging for breast cancer, we calculated the diagnostic yield (DY) of staging chest CT examinations. DY refers to the likelihood that a test will provide the information needed to establish a diagnosis and was defined as the proportion of patients with true-positive metastases among all patients (number of true-metastases divided by total number of patients). Also, we evaluated the false-referral rate (FRR) of staging chest CT. FRR refers to the likelihood that a test will provide false information necessitating additional tests, and it was defined as the proportion of patients with false-positive metastases among all patients (number of false-positive results divided by total number of patients) [28,29]. The associations between categorical variables and the rate of initial metastasis were evaluated using the χ2 test or Fisher’s exact test, as appropriate. A *p*-value < 0.05 was considered to indicate statistical significance. SPSS version 25.0 software (SPSS, Chicago, IL, USA) was used for statistical analysis.

## 3. Results

The characteristics of the 840 patients are summarized in Table 1. The mean age was 50.0 years (range, 27–80 years). The number of cancers in clinical stages 0/I, II, III, and IV were 457 (53.5%), 298 (34.9%), 92 (10.8%), and 8 (0.9%), respectively. The histopathologic types of breast cancers are summarized in Table 2. Molecular subtype was evaluated in 841 breast cancers. There were 709 breast cancers of luminal type (84.3%), 55 cases of HER2-enriched type (6.5%), and 77 breast cancers of TN type (9.2%). The distribution of clinical stage according to molecular subtype is summarized in Table 3.

Of the 855 breast cancers (including 15 bilateral breast cancers), 20 (2.3%) were proven as having initial distant metastases (Figure 1). None of the patients with bilateral breast cancer showed initial metastasis, and we regarded the bilateral tumors as two individual breast cancers. Of the 20 breast cancers with distant metastases, 19 involved invasive ductal carcinomas, while the last case involved clinical stage II sarcoma.

Of the 20 breast cancers with initial distant metastases, 18 metastases were detected through staging chest CT while two metastases were not identified by staging chest CT (one skeletal metastasis could not be identified even in a retrospective review, and one pulmonary metastasis was considered as a benign nodule on staging chest CT).

Eighteen cases (18/20) of metastases were detected on staging chest CT. The DYs for clinical stage 0/I, II, and III cases were 0.2%, 1.7%, and 4.3%, respectively (Table 4). Among the 457 clinical stage 0/I patients, one distant metastasis (lung metastasis) was detected on chest CT. Among the 298 clinical stage II patients, five distant metastases (two cases in lung, two cases in bone, and one case in the mediastinal LN) were detected through staging chest CT. Among the 92 clinical stage III patients, four distant metastases (one case in bone, one case in mediastinal LN, one case in lung and one case in liver) were detected on chest CT (Figure 2). In initial breast MRI, distant metastases in bone and mediastinal LN were detected in the eight clinical stage IV patients, and additional bone, lung, and liver metastases were identified through staging chest CT. In clinical stage IV patients, sites with distant metastasis were mediastinal LN (six cases), bone (five cases), lung (four cases), and liver (one case).

Of the 841 breast cancers with known molecular subtype, there were 14 metastases of luminal subtype, two metastasis of HER2-enriched subtype, and two metastases of the TN subtype, respectively. The DYs of the luminal subtype, HER2-enriched subtype, and TN subtype were 1.7%, 3.6%, and 2.6%, respectively (Table 4). Among the 12 patients with distant metastases in clinical stage 0/I–III, 10 cases were of the luminal type, one case was of the HER2-enriched type, and one case was of the unknown type. The cases with distant metastases in the TN type were all initial cIV stage. The proportions of metastasis based on clinical stage and molecular subtype are depicted in Table 5, Figure 1 and Figure 3.

DY of staging chest CT for the detection of distant metastasis was significantly associated with clinical stage (*p* = 0.000). However, molecular subtype of breast cancer did not affect the proportion of initial distant metastasis, unlike clinical stage. There was no statistical difference in the DYs of staging chest CT for the detection of metastasis based on molecular subtype (*p* = 0.343). Histologic grade and nuclear grade were evaluated and identified in 751 patients. Histologic grade and nuclear grade were not significantly associated with the DYs (*p* = 0.399, *p* = 0.225, respectively) (Table 6).

The overall FRR was 2.2% (19/855). When stratified by clinical stage, the FRRs in clinical stages 0/I, II, III and IV were 2.8% (13/457), 1.3% (4/298), 2.2% (2/92), and 0.0% (0/8), respectively (Table 4). There was no association between clinical stage and FRR (*p* = 0.526). And also, molecular subtype was not related to FRR (*p* = 0.481) (Table 6).

Among 840 patients, ancillary findings that needed follow-up or further treatment were identified in 48 patients (5.7%) (Table 7). A total of 273 patients showed a pulmonary nodule (32.5%, 273/840). Among them, 11 cases of pulmonary metastases were confirmed (4%,11/273) and primary lung cancer was incidentally detected in six patients (2.2%, 6/273). All these patients were non-smokers and all lung cancers were adenocarcinomas. Synchronous pancreatic cancer, thyroid cancer, and thymoma were detected in one case each. Subsolid pulmonary nodules larger than 6 mm were identified in 19 patients (2.3%, 19/840).

## 4. Discussion

The molecular subtype of breast cancer is an important prognostic factor. Luminal-type cancers show lower locoregional recurrence and metastasis rates. Luminal-type cancers are known to show better prognosis than HER2-enriched tumors or TN tumors [15,16,18]. Initial distant metastasis in newly diagnosed breast cancer patients is another important prognostic factor. However, several studies reported that staging chest CT in early-stage breast cancer patients was not recommended due to the low prevalence of distant metastasis [4,5,30,31]. Therefore, recent guidelines suggest that staging chest CT of breast cancer is recommended for higher clinical stages of breast cancer.

Although the molecular subtype is an important prognostic factor for breast cancer, no previous studies have assessed the usefulness of staging chest CT based on molecular subtypes. Our results showed that the DY of staging chest CT showed no significant difference in relation to the molecular subtype of breast cancer. Therefore, the molecular subtype of breast cancer does not provide additional information to determine whether preoperative staging chest CT should be performed. In addition, four of six patients who showed initial metastasis in clinical stage II had luminal-type cancers, which are known to show better prognosis.

The guidelines recommend that chest CT should be performed only when clinically suspected, but this recommendation is somewhat ambiguous [6,7,8]. Therefore, preoperative staging chest CT is still performed in many cases for various reasons [9,11,32]. To assess the discrepancy between the guidelines and clinical practice, we estimated the benefits of staging chest CT based on clinical stage by evaluating the DY. The DYs of chest CT for detection of distant metastasis were 0.2% in clinical stage 0/I breast cancer, 1.7% in clinical stage II breast cancer, and 4.3% in clinical stage III breast cancer. These results were similar to those reported in previous studies [5,30,31].

The preferential site of metastasis is known to differ based on the molecular subtype. Luminal-type cancers metastasize into bone more frequently, while hormone receptor-negative cancers preferentially metastasize into visceral organs [33]. In our study, only 35.7% (5/14) of luminal cancers with initial metastasis showed bone metastasis. On the other hand, 100.0% (4/4) of hormone receptor-negative cancers with initial metastasis showed metastasis to visceral organs such as lung, liver, and lymph node. This result was different from the previously reported findings, but the differences may be attributed to the bias introduced by the small sample size.

We assessed ancillary findings other than metastasis in preoperative staging chest CT. There were 256 (30.5%) cases of indeterminate noncalcified pulmonary nodules in our study. There has been no consensus about how to manage the indeterminate pulmonary nodules in patients with underlying malignancy [34]. Brothers et al. reported that indeterminate pulmonary nodules could be associated with disease recurrence, however the incidence was very low (1.3%, 1/73) in their study [35]. Further study will be required to assess the association between indeterminate pulmonary nodules and the likelihood of malignancy and to establish the guidelines for management of indeterminate nodules in patients with malignancy.

There are several potential factors that influence physicians to perform staging chest CT, including stage, aggressive tumor biology, inflammatory breast cancer, and so on [11]. Our study has shown that molecular subtype did not provide information indicating the need for initial staging chest CT. However, there were considerably smaller numbers of cases of the HER2-enriched and TN type tumors than the luminal-type tumors in our study. Therefore, a larger study including large number of the HER2-enriched and TN type tumors is needed. Also, further research about how other factors are related to the DY of chest CT in detecting metastasis will be required.

Our study had several limitations. First, the small number of patients has limited power, and further studies with larger samples are needed. Second, there were differences in the percentage of tumor subtypes. Since the number of cases with the HER2-enriched and TN types were small, there might be bias. Third, chest CT examinations were not done in all patients, especially in early-stage breast cancer patients. Therefore, the percentage of initial metastasis might have been overestimated or underestimated.

## 5. Conclusions

The molecular subtype of breast cancer could not provide useful information to determine the need for a preoperative chest CT examination for detection of distant metastasis in patients with newly diagnosed breast cancer. Preoperative chest CT should be considered in advanced breast cancer patients with clinical stage III and IV disease, regardless of molecular subtype.

## Figures and Tables

**Figure 1 jcm-10-00906-f001:**
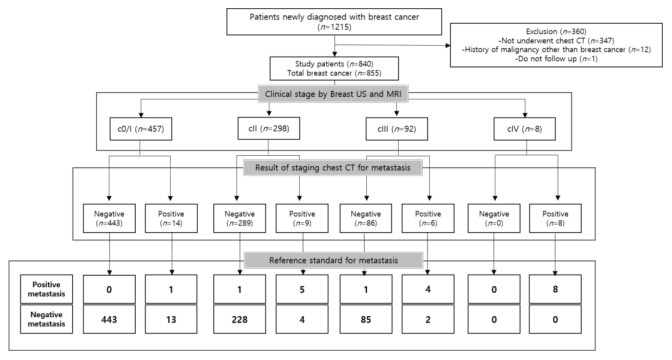
Patient flow diagram showing reference standard for metastasis based on clinical stage.

**Figure 2 jcm-10-00906-f002:**
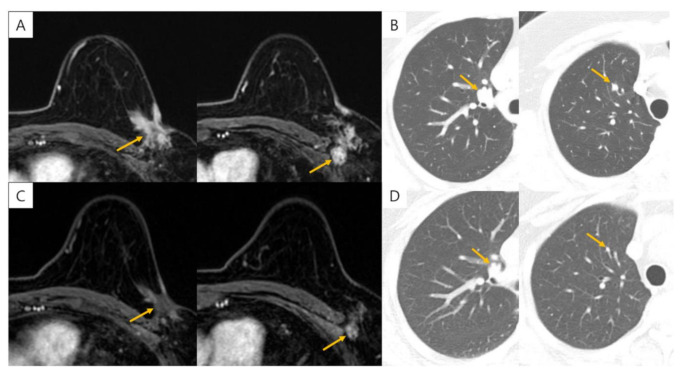
56-year-old female with breast cancer in Lt breast. (**A**) On initial MRI, there was about a 3.5 cm irregular shape enhancing mass in her Lt breast with skin invasion and multiple metastatic lymph nodes in Lt axilla level I and II. Initial clinical stage was III based on MRI. (**B**) Multiple metastatic pulmonary nodules were detected on staging chest CT and clinical stage was upstaged from stage III to IV. (**C**) After neoadjuvant chemotherapy, primary breast mass and axillary lymph nodes decreased in size. (**D**) Pulmonary metastatic nodules also decreased in size after neoadjuvant chemotherapy, consistent with the response of primary lesion.

**Figure 3 jcm-10-00906-f003:**
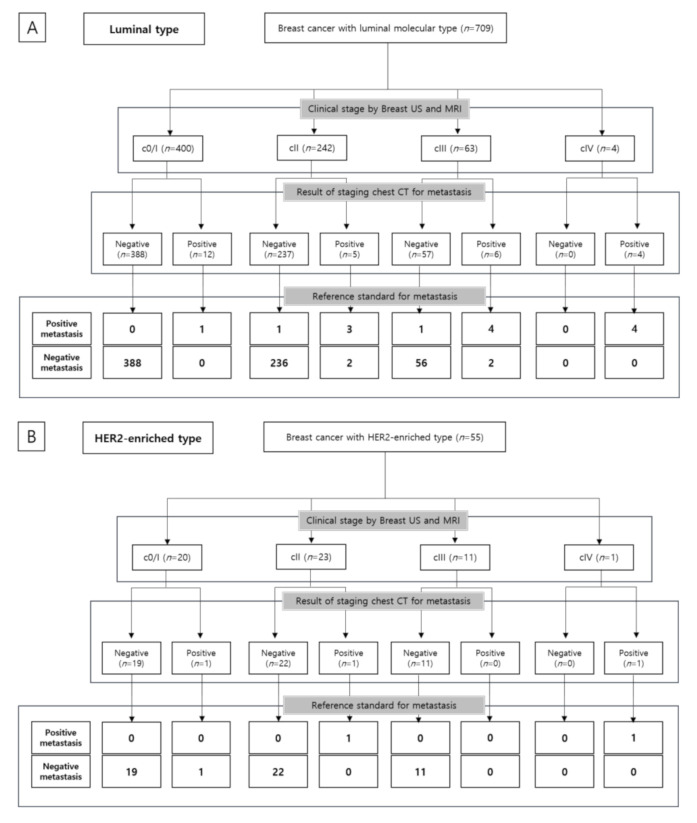
Patient flow diagram showing reference standard for metastasis based on molecular subtype. (**A**) Luminal type; (**B**) HER2-enriched type; (**C**) Triple-negative type.

**Table 1 jcm-10-00906-t001:** Clinicopathologic characteristics of the 840 patients with 855 breast cancers.

Characteristics	Number of Cancer (%)
Age (years)	840 patients
≤35	35
36–50	430
≥51	375
Clinical stage	855 cases
0	28 (3.3)
I	429 (50.2)
II	298 (34.9)
III	92 (10.8)
IV	8 (0.9)
ER status	855 cases
Positive	679 (79.4)
Negative	162 (18.9)
Unknown	14 (1.6)
PgR status	855 cases
Positive	632 (73.9)
Negative	206 (24.1)
Unknown	17 (2.0)
HER2 amplification	855 cases
Positive	169 (19.8)
Negative	665 (77.8)
Unknown	21 (2.5)
Molecular type	855 cases
Luminal	709 (82.9)
HER2-enriched	55 (6.4)
Triple negative	77 (9.0)
Unknown	14 (1.6)
Histologic grade	
High	246 (28.8)
Low	505 (59.1)
Unknown	104 (12.2)
Nuclear grade	
High	301 (35.2)
Low	450 (52.6)
Unknown	104 (12.2)

ER: estrogen receptor, PgR: progesterone receptor, HER2: human epidermal growth factor receptor 2.

**Table 2 jcm-10-00906-t002:** Histologic type of breast cancer.

Histology	Number of Cancer (*n* = 855)
Invasive ductal carcinoma	759
Invasive lobular carcinoma	49
Mucinous carcinoma	19
Metaplastic carcinoma	7
Tubular carcinoma	6
Micropapillary carcinoma	5
Others	10

**Table 3 jcm-10-00906-t003:** Distribution of clinical stage according to molecular subtype (*n* = 841).

		Clinical Stage by US and MRI				
		0/I	II	III	IV	Total
Molecular subtype	Luminal	400 (56.4%)	242 (34.1%)	63 (8.9%)	4 (0.6%)	709
	HER2-enriched	20 (36.4%)	23 (41.8%)	11 (20.0%)	1 (1.8%)	55
	Triple negative	30 (39.0%)	29 (37.7%)	16 (20.8%)	2 (2.6%)	77
Total		450	294	90	7	841

US: ultrasound, MRI: magnetic resonance imaging, HER2: human epidermal growth factor receptor 2.

**Table 4 jcm-10-00906-t004:** Diagnostic yield and false-referral rate of chest CT for the detection of distant metastasis according to clinical stage and molecular subtype.

**Clinical Stage (*n* = 855)**	**Diagnostic Yield**	**False-Referral Rate**
0/1	0.2% [1/457]	2.8% [13/457]
II	1.7% [5/298]	1.3% [4/298]
III	4.3% [4/92]	2.2% [2/92]
IV	100.0% [8/8]	0.0% [0/8]
**Molecular Subtype (*n* = 841)**	**Diagnostic Yield**	**False-Referral Rate**
Luminal	1.7% [12/709]	2.1% [15/709]
HER2-enriched	3.6% [2/55]	1.8% [1/55]
Triple negative	2.6% [2/77]	3.9% [3/77]
Unknown	[2/14]	[0/14]

HER2: human epidermal growth factor receptor 2.

**Table 5 jcm-10-00906-t005:** The proportion of initial metastasis based on clinical stage and molecular subtype.

	Clinical Stage			
Molecular Subtype	0/I	II	III	IV
Luminal	0%	1.65%	6.30%	100%
	(0/400)	(4/242)	(4/63)	(4/4)
HER2-enriched	0%	4.30%	0%	100%
	(0/20)	(1/23)	(0/11)	(1/1)
Triple negative	0%	0%	0%	100%
	(0/30)	(0/29)	(0/16)	(2/2)

HER2: human epidermal growth factor receptor 2.

**Table 6 jcm-10-00906-t006:** Factors influencing diagnostic yield and false-referral rate.

	Diagnostic Yield			False-Referral Rate		
Factors	Negative Metastasis	Positive Metastasis	*p-*Value	Negative Finding	False Positive	*p-*Value
Clinical stage			0.000 *			0.526
0/I	456	1		444	13	
II	293	5		294	4	
III	88	4		90	2	
IV	0	8		8	0	
Age			0.063			0.808
≤35	35	0		34	1	
36–50	430	5		427	8	
≥51	372	13		375	10	
Molecular subtype			0.343			0.481
Luminal	697	12		694	15	
HER2-enriched	53	2		54	1	
TN	75	2		74	3	
Histologic grade			0.399			0.130
High	243	3		237	9	
Low	502	3		496	9	
Nuclear grade			0.225			0.027 *
High	297	4		289	12	
Low	448	2		444	6	

* Statistical significance below 0.05; HER2: human epidermal growth factor receptor 2, TN: triple negative.

**Table 7 jcm-10-00906-t007:** Ancillary findings at staging contrast-enhanced chest CT.

	**Total (*n* = 840)**	**c0/I** **(*n* = 445)**	**cII** **(*n* = 295)**	**cIII** **(*n* = 92)**	**cIV** **(*n* = 8)**
Lung cancer	6 (0.7%)	3 (0.7%)	2 (0.7%)	1 (1.1%)	0 (0%)
Cancer *	3 (0.4%)	1 (0.2%)	1 (0.3%)	1 (1.1%)	0 (0%)
Subsolid nodule **	19 (2.3%)	12 (2.7%)	4 (1.4%)	3 (3.3%)	0 (0%)
Tuberculosis/Non- tuberculous mycobacterium	4 (0.5%)	3 (0.7%)	0 (0%)	1 (1.1%)	0 (0%)
Pneumonia	4 (0.5%)	0 (0%)	2 (0.7%)	2 (2.2%)	0 (0%)
Others ***	12 (1.4%)	8 (1.8%)	4 (1.4%)	0 (0%)	0 (0%)
Total	48 (5.7%)	27 (6.1%)	13 (4.4%)	8 (8.7%)	0 (0%)

* Cancers include pancreas, thyroid cancer and thymoma. These lesions confirmed by surgical resection; ** Subsolid nodule that was larger than 6 mm; *** Others included pancreas cystic mass, anterior mediastinal cystic mass, sarcoidosis, pulmonary arteriovenous malformation and pulmonary thromboembolism.

## Data Availability

The data presented in this study are available on request from the corresponding author. The data are not publicly available due to ethical restrictions.

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
