# Peer review of "Usefulness of Staging Chest CT in Breast Cancer: Evaluating Diagnostic Yield of Chest CT According to the Molecular Subtype and Clinical Stage"

_jcm, 2021, doi:10.3390/jcm10050906_

Round 1

Reviewer 1 Report

I consider this article very interesting, which assesses the efficacy of diagnostic tests in the staging of breast cancer that do not provide any benefit. We could have the doubt as to whether the different phenotypes would condition the validity of these examinations, although in this case the number of cases of the TN and HER2 phenotype is considerably lower than the luminal one, it can be concluded that even in these phenotypes they do not provide any benefit either.

The only thing I would add is if the study could be done excluding cases in stages IV, which could provide even better data to conclude that a thoracic CT scan is not necessary in the initial stages of breast cancer.

Author Response

Thank you for your comment. As you comment, it is not very recommended to use chest CT for detection of distant metastasis in early stage of breast cancer. However, we found additional metastatic lesions which were not discovered in breast MRI and US through chest CT in stage IV cases. Therefore, staging chest CT could be beneficial in stage IV breast cancer.

Reviewer 2 Report

JCM-1112978
To the Authors

Dear Authors, I have received your manuscript entitled "Usefulness of staging chest CT in breast cancer: evaluating di-2 agnostic yield of chest CT according to the molecular subtype 3 and clinical stage". I deem this to be a very well-written and methodologically sound study, and I have only two minor comments that might be considered to improve the presentation of your interesting results.

1) In the Introduction, you might consider to better clarify the logic transition between the first section (up to line 44) and the section on molecular subtyping (line 45 onwards), for example linking the two sections by expanding the concept you present very briefly on the importance of "tumor biology" as a driver of chest CT recommendation in the initial stage of breast cancer. This would "smoothen" the logical flow of your Introduction. A potential solution would be to anticipate the paragraph reporting Simos et al. survey, then mentioning it again to introduce the absence of clear reports examining the DY of chest CT according to molecular subtype.

2) In the Discussion, I would expect a brief comment to clinically contextualize the paragraph between lines 273 and 278 (now it is more a re-statement of results), also referring to relevant literature. I would also suggest you briefly mention, before the limitations, what further research pathways could stem from your study, for example hypothesizing a larger study examining the real impact and/or interplay of the other "drivers" of chest CT referral for breast cancer staging you have mentioned in the Introduction.

Thank you and best regards
